# Concurrence of anemia and stunting and associated factors among children aged 6 to 59 months in Peru

**Alessandra Rivera**●, **Víctor Marín**⊙*●, **Franco Romaní**⊙

Faculty of Human Medicine, Universidad de Piura, Lima, Peru

● These authors contributed equally to this work.

* victor.marin@alum.udep.edu.pe

**Data Availability Statement:** We are submitting the raw data from our analysis in DTA format, and these files are included as supporting information. The file is named S1 Data.

## Abstract

Anemia and stunting are two health problems in the child population; therefore, their concurrence needs to be quantified. We estimated the prevalence of concurrent anemia and stunting (CAS) in children aged 6–59 months and identified the factors associated with this condition. The data came from the Demographic and Health Survey of Peru (DHS), 2022. The study design was cross-sectional and included 19,191 children. Height and hemoglobin measurement followed the specifications of National Health Institute of Peru. To reduce error in measures, the anthropometry personnel was training, the quality of measuring equipment was ensuring, and protocolized techniques and procedures was applying. Hemoglobin concentration was measured in capillary blood using the Hemocue model Hb 201+. Stunting was defined as a height-for-age Z-score less than minus two standard deviations (SD) from the median, following the 2006 WHO child growth standard. Anemia was classified into mild (10.0 to 10.9 g/dL), moderate (7.0 to 9.9 g/dL), severe (< 7.0 g/dL), and no anemia (11.0 to 14.0 g/dL). We performed a bivariate analysis to evaluate factors associated with CAS. To include variables in the multivariate analysis, we applied a statistical criterion (p < 0.10 in the crude analysis) and an epidemiological criterion. We used a binary logistic hierarchical regression model. The prevalence of CAS was 5.6% (95%CI: 5.2 to 5.9). The modifiable factors associated with higher odds of CAS were: "poorest" (aOR: 3.87, 95%CI: 1.99 to 7.5) and "poorer" (aOR: 2.07, 95%CI: 1.08 to 3.98) wealth quintiles, mother with no formal education or primary (aOR: 2.03, 95%CI: 1.46 to 2. 81), father with no formal education or primary (aOR: 1.55, 95%CI: 1.16 to 2.07), no improved water source (aOR: 1.36, 95%CI: 1.10 to 1.68), no roof with improved material (aOR: 1.49, 95%CI: 1.12 to 1.98) and low birth weight (aOR: 7.31, 95%CI: 4.26 to 12.54). In Peru, five out of every 100 children suffer from anemia and stunting simultaneously; there are modifiable factors that, if addressed, could reduce their prevalence.

## Introduction

Anemia is a prevalent medical condition in low- and middle-income countries, especially in Latin America and the Caribbean [1, 2]. Although its magnitude has decreased in recent years,

**Funding:** The authors received no specific funding for this work.

**Competing interests:** The authors have declared that no competing interests exist.

it is still the main cause of years lived with disability among children [3] and is a risk factor for all-cause mortality in this population [4]. Peru has one of the highest prevalences of childhood anemia, which represents a persistent challenge for its health system [5]. In the year 2022, in Peru there was an increase from 38.8% to 42.4% in the prevalence of anemia in children aged 6 to 35 months; with this increase, in rural areas anemia affected 51.5% of children [6].

In addition to anemia, stunting in children poses a persistent public health challenge in Peru [6]. Despite the decrease in the prevalence of stunting in children under five years in the country, which went from 14.4% to 11.7% between 2015 and 2020, it is important to note that inequalities in its distribution have been accentuated, disproportionately affecting rural areas and the most vulnerable and poorest populations in Peru [6].

While it is true that the coexistence of stunting and overweight is the most frequent representative of the double burden of malnutrition, the concurrence of anemia and stunting in the infant population is another relevant manifestation of this phenomenon. Coexistence of these conditions is not uncommon among children aged 6–59 months. For example, in Ethiopia, the prevalence was 24.8% [7], while in Ghana, it was 11.8% [8]. Due to its magnitude, this problem is considered an emerging challenge for public health, given the complications derived from the interaction between both conditions [9].

Stunting is known to cause increased morbidity and premature mortality, impaired cognitive function, lower school performance, stigma, discrimination, as well as increased probability of developing chronic diseases in the future [10, 11]. Although studies have been conducted in Peru to estimate the prevalence of anemia and stunting separately, there are still no estimates of the prevalence of the concurrence of both conditions in this country.

For the study of the determinants of stunting, hierarchical conceptual frameworks have been adopted, in which distal determinants —including socioeconomic factors— indirectly affect the occurrence of stunting through their influence on intermediate determinants, such as feeding practices, access to health care, household conditions and hygiene [12, 13]. In addition, they also influence proximal determinants, which include maternal characteristics, the child's birth weight, and the child's dietary diversity. In this context, we propose that the use of this approach will help to identify and evaluate the factors that contribute to the presence of concurrent anemia and stunting in preschool children in Peru [6].

Among the objectives of UNICEF's Nutrition Strategy for 2020–2023 is to prevent undernutrition, micronutrient deficiencies and overweight during the first five years of life [14]. While UNICEF has a conceptual framework for preventing all forms of malnutrition, it does not identify specific risk or protective factors for the concurrent anemia and stunting (CAS). The absence of a specific conceptual framework for this concurrence is because there are few studies that have explored the factors associated with the CAS. To our knowledge, factors associated with the CAS have not been examined in a nationally representative sample of children under five years of age in Peru.

Given that anemia and stunting are health problems with common determinants in Peru, it is plausible to propose the concurrence of both diseases in preschoolers. Another hypothesis is that CAS responds to a series of determinants that could be explored through a hierarchical model [15]. Therefore, the aim of this study was to determine the prevalence of concurrent anemia and stunting (CAS) in children 6 to 59 months of age and its association with sociodemographic factors.

## Methods

### Study design

An analytical observational study of secondary sources was conducted. The source of information was data from the 2022 Demographic and Family Health Survey (DHS) conducted in

Peru by the National Institute of Statistics and Informatics (INEI). The DHS collected the information through face-to-face interviews conducted in selected households during the months of January to December 2022.

For the fieldwork, the INEI used a computer application implemented on tablets, data collection was performed by an interviewer and an anthropometrist, the latter was responsible for anthropometric and hemoglobin measurement in the target population [16]. The personnel described were mainly health professionals.

The DHS 2022 sample is the product of a two-stage, probabilistic, balanced, stratified and independent design at the departmental level and by rural and urban area. For 2022, it was planned to interview 36,760 households in which 43,257 women aged 12 to 49 years and 26,389 children aged 59 months or younger were to be studied. The number of households in which interviews were carried out was 35,287, with 35,787 complete interviews of women between the ages of 12 and 49.

The selected sample provides representative estimates of the population of Peru, its 24 departments and the Constitutional Province of Callao, and its natural regions (Metropolitan Lima, Rest of Coast, Highlands, and Jungle) [6].

## Setting and participants

The DHS has as its target population the usual residents and those who, not being residents, stayed overnight in the dwelling the night before the day of the interview. The survey encompasses women from 12 to 49 years old and children under 5 years old, all children under 12 years old and one person 15 years old or older in each selected household. The DHS conducts interviews with all women in the selected household.

Estimates for this analysis were made in a subpopulation defined by the following inclusion criteria: a. children aged 6 to 59 months, b. children with anthropometric height/age data, c. children with mothers aged 15 to 49 years, and d. children with hemoglobin data. Given the possibility of women with more than one child ≤ 59 months at the time of the interview, it was decided to include the last live birth in the present analysis. In this way we avoided having correlated observations between siblings of the same mother.

## Procedures for anthropometry and hemoglobin measurement

The DHS aimed to reduce the margin of error in the height and hemoglobin measurement through the following measures: a. Training of anthropometry personnel according to national technical standards; b. Ensuring the quality of measuring equipment; c. Protocolizing techniques and procedures; and d. Ensuring the quality of data recording [17].

For quality assurance of the measurements, the DHS supervisors conducted direct observation of the anthropometric and hemoglobin measurement technique by the anthropometric personnel. The evaluation of intraobserver reliability of the hemoglobin measurement was considered adequate when the staff measurement had differences of less than 0.5 g/dL with respect to a first measurement. For height, interobserver reliability was assessed by a double measurement and was considered adequate when there was no difference of more than +/- 3 mm between the staff measurement and the supervisor's measurement. Both reliability assessments were performed in a pilot fieldwork [17].

The procedure for measuring the weight of children aged 6 months to under 2 years and those aged 2 years and older is detailed in the national technical standard [18]. Weight measurements were recorded in kilograms, with precision to one decimal place and no rounding. Wooden, mobile, multipurpose height measuring devices were used to measure height according to the technical specifications of INS-CENAN and UNICEF [19]. To measure length in

children under 2 years old, the measuring rod was placed in a horizontal position on a rigid, flat surface and then continued according to the procedure described [18]. To measure height in children aged 2 years and older, the measuring rod was placed on a solid surface, such as a wall or table, to ensure its stability. The height value was recorded and verified immediately afterwards by dictating each datum.

Hemoglobin concentration was measured in capillary blood. The Hemocue model Hb 201 + portable device was used, which is based on the modified azidametahemoglobin reaction. The HemoCue Hb is calibrated with the international reference method for hemoglobin determination (ICSH3). In children aged 12 to 71 months the puncture was performed in the center of the middle fingertip of the hand; in children younger than 12 months the peripheral puncture of the heel was performed, according to described protocol [18]. The anthropometrist dictated the hemoglobin result to one decimal place, which was recorded.

## Variable definition

Stunting was defined when the child had a height-for-age Z-score less than minus two (-2) standard deviations (SD) below the median according to the 2006 WHO child growth standard (variable HW70) [20]. The measurement categories were not stunting (HW70 $\geq$ -2 SD), moderate stunting (-3 SD $\geq$ HW70 < -2 SD) and severe stunting (HW70 < -3 SD). For definition of concurrent presentation of anemia and stunting (CAS), this variable was recategorized as stunting (including moderate and severe form) and not stunting.

Hemoglobin concentrations were categorized into normal (11.0 to 14.0 g/dL), mild (10.0 to 10.9 g/dL), moderate (7.0 to 9.9 g/dL) and severe (< 7.0 g/dL) anemia. These categories considered the "altitude-adjusted hemoglobin level". For bivariate and multivariate analysis, anemia was defined as present (hemoglobin < 11.0 g/dL) or absent ($\geq$ 11.0 g/dL).

For the present study, the variable of interest was the CAS which was defined when the child had simultaneous presence of anemia and stunting. The measurement of this variable was dichotomous (presence or absence).

The variables included were grouped into distal, intermediate, and proximal factors, as has been done in other studies [12, 13]. The proximal factors were child sex, child age [13], child-birth weight, type of delivery, cesarean delivery, minimum dietary diversity [21] and immediate breastfeeding [21].

The variables considered intermediate factors were source of water in the home [21], ownership of electrical appliances [21], means of transportation [21], floor material, wall material, and roof material of the home [21]. Finally, distal factors were area of residence, natural region, mother currently working, wealth index, maternal marital status, maternal insurance coverage, maternal education, paternal education, and maternal ethnic group membership. These variables were selected based on UNICEF's conceptual framework on the determinants of maternal and child nutrition [15], which has been used in previous studies [22–24] (S1 Table).

## Statistical analysis

We considered the complex sample design of the DHS 2022 and used the variable HV001 (cluster), HV022 (stratum), the sample weight was calculated by dividing the variable HV005 (household weight factor) by 1,000,000. We performed the analysis for complex samples using the *svyset* command in STATA version 16.0. Estimates were made for the subpopulation defined by children aged 6 to 59 months who had complete measurements for the main variables, whose mothers were aged 15 to 49 years, and with respect to the last live birth.

We performed the descriptive analysis of anemia and stunting, as well as the concurrence of both by estimating the weighted point prevalence and its respective 95% confidence interval

(95%CI). These estimates were also made for the study covariates. We applied a bivariate analysis to evaluate the factors associated with CAS, for this purpose, the comparison of proportions was performed using Pearson's Chi-square test with Rao and Scott's second-order correction [25]. In addition, we estimated the odds ratio with its 95%CI with a bivariate binary logistic regression.

For the multivariate analysis we applied a three-level binary hierarchical logistic regression model. The conceptual framework of stunting and CAS supports the use of this analytical approach, as the explanatory variables are configured into distal, intermediate, and immediate factors. The adaptation of a hierarchical model to study the determinants of CAS is based on the recommendations of Victora et al. [26]. The epidemiological criterion allowed the identification of the explanatory variables, then based on a statistical criterion, those variables with a p value < 0.10 in the crude analysis were included for the multivariate analysis. The choice of a 0.10 threshold was made in consideration of the theoretical model guiding the identification of variables. This cutoff was selected to afford a more sensitive criterion while simultaneously avoiding an undue increase in the risk of type I error.

The model 1 included the distal factors that reached the required p value. Model 2 included the intermediate and distal factors that met the statistical criterion in the bivariate analysis. Subsequently, model 3 included the immediate, intermediate, and distal factors that met the statistical criterion in the bivariate analysis. The statistical significance of each explanatory variable in the models was defined with a value of p < 0.05. Specifically, the p-value obtained for the variable in the first model in which it was entered was considered, while estimates of statistical significance in successive models were not taken into account [21, 26].

This approach, in the first model, allows estimation of the strength of association of the distal variables and the outcome without considering adjustment for mediating variables, which are not confounders. In the second model, the underlying or intermediate variables are adjusted for the distal variables (confounders), while the remaining effect of the distal variables reflects the part not mediated through the underlying variables. Finally, model 3 presents the effect of the proximal variables in the presence of the distal and underlying confounders [26].

Binary hierarchical logistic regression allowed estimation of the adjusted odds ratio and its respective 95%CI. The diagnosis of multicollinearity among the explanatory variables was performed using the standard errors of the regression coefficients; values greater than 2.0 were indicative of multicollinearity [25]. The goodness of fit of the models was estimated with pseudo R2 of Nagelkerke and McFadden. The latter analysis was performed in SPSS version 25. The rest of the statistical analysis was performed with STATA version 16.

### Ethics statements

The information analyzed is publicly available on the web portal of the National Institute of Statistics and Informatics (https://proyectos.inei.gob.pe/microdatos/). As a secondary source study, the researchers did not intervene or have direct contact with the children. However, the primary survey obtained informed consent from the parents of the participating children. This analysis was performed using an anonymized database. The study's research protocol was approved by the Institutional Research Ethics Committee of the Universidad de Piura.

## Results

### Participant selection

After merging the databases, we identified 22 611 records of children under 5 years of age and their mothers. We excluded children aged 5 months or less (n = 1974), children without anthropometry data (n = 324), those whose mothers were 12–14 years old (n = 6), and children

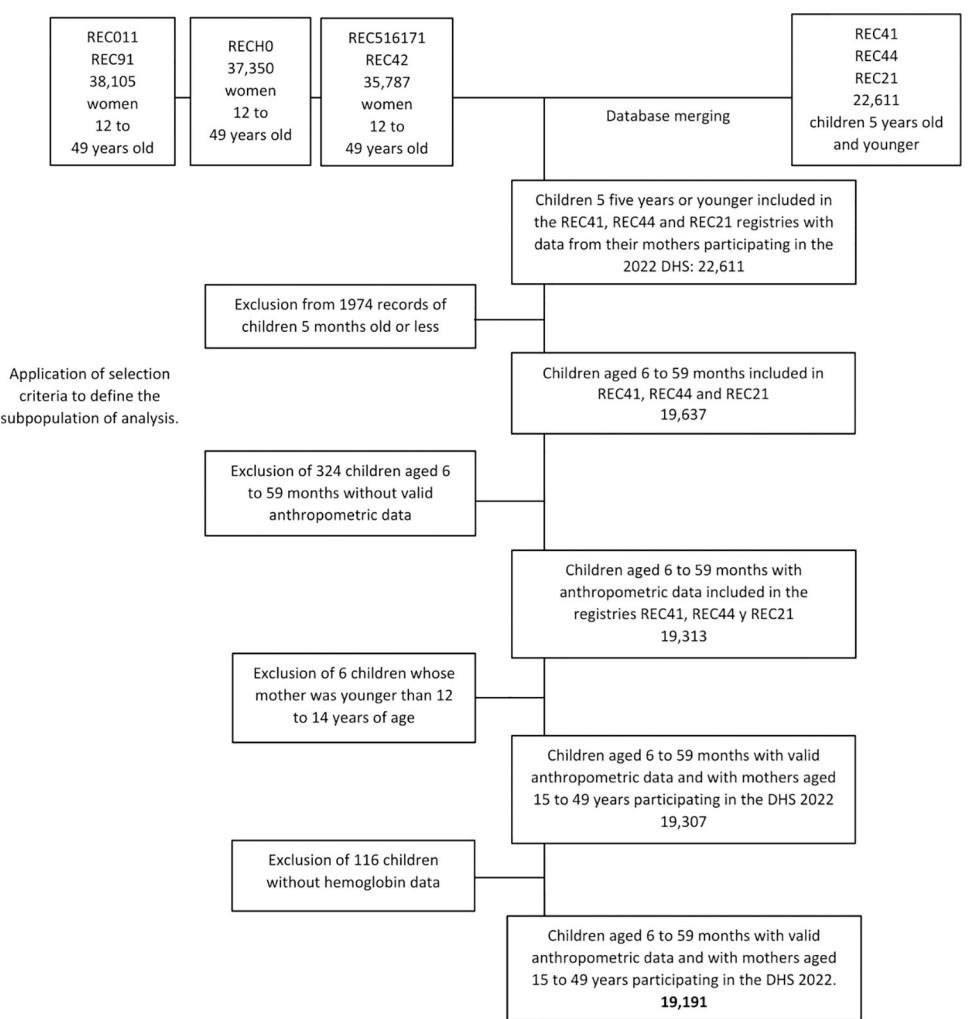

**Fig 1. Flow chart for the selection of participants in the analysis.** DHS: Demographic and Family Health Survey.

without hemoglobin data (n = 116). Finally, 19 191 children met the inclusion criteria, and their data were analyzed (Fig 1).

## Sample characteristics

Among the proximal factors, 51.5% were boys and 85.9% had a normal birth weight. According to the mother's report, 81.3% reached a minimum dietary diversity. Regarding the underlying factors, more than 90% had a water source, improved wall and roof material, and household appliances. Among the distal factors, 73.1% were from urban areas, 82.4% of the mothers had some type of union; with respect to educational level, 33.3% of the mothers had higher education, a percentage like that of the male couple (33.4%). Other characteristics of the sample are presented in Table 1.

## Prevalence of concurrence of anemia and stunting

The prevalence of concurrence of anemia and stunting in children aged 6 to 59 months was 5.6% (95%CI: 5.2 to 5.9). The prevalences of anemia and stunting separately are shown in

**Table 1. Frequency of sociodemographic factors associated with the concurrence of anemia and stunting in children aged 6–59 months (n = 19 191).**

| Variables | Unweighted absolute frequency | Weighted proportion | 95% confidence interval | |
|---|---|---|---|---|
| | | | LL | UL |
| **Basic factors (distal)** | | | | |
| Residence | | | | |
| Urban | 13,057 | 73.1 | 72.8 | 73.4 |
| Rural | 6,134 | 26.9 | 26.6 | 27.2 |
| Mother currently working | | | | |
| No | 8,182 | 42.5 | 41.7 | 43.4 |
| Yes | 11,009 | 57.5 | 56.6 | 58.3 |
| Wealth index | | | | |
| Poorest | 5,993 | 27.3 | 26.7 | 27.8 |
| Poorer | 5,175 | 24.5 | 23.8 | 25.2 |
| Medium | 3,778 | 20.2 | 19.5 | 20.9 |
| Wealthier | 2,593 | 16.3 | 15.6 | 16.9 |
| Wealthiest | 1,652 | 11.9 | 11.3 | 12.5 |
| Maternal marital status | | | | |
| United | 15,855 | 82.4 | 81.7 | 83.0 |
| Not united | 3,336 | 17.6 | 17.0 | 18.3 |
| Maternal insurance coverage | | | | |
| No | 1,852 | 11.2 | 10.6 | 11.8 |
| Yes | 17,339 | 88.8 | 88.2 | 89.4 |
| Maternal education | | | | |
| No formal education/primary | 3,631 | 18.3 | 17.7 | 18.8 |
| Secondary | 9,514 | 48.5 | 47.6 | 49.4 |
| Higher | 6,046 | 33.3 | 32.5 | 34.0 |
| Paternal education (n = 18,304) | | | | |
| No formal education/primary | 2,697 | 14.4 | 13.9 | 15.0 |
| Secondary | 9,702 | 52.2 | 51.3 | 53.0 |
| Higher | 5,905 | 33.4 | 32.6 | 34.2 |
| Maternal ethnicity | | | | |
| No | 17,179 | 92.2 | 91.9 | 92.5 |
| Yes | 2,012 | 7.8 | 7.5 | 8.1 |
| Natural region | | | | |
| Metropolitan Lima | 2,310 | 24.9 | 24.6 | 25.2 |
| Rest of coast | 5,504 | 27.4 | 27.0 | 27.8 |
| Highlands | 6,383 | 28.7 | 28.2 | 29.2 |
| Jungle | 4,994 | 19.1 | 18.7 | 19.4 |
| **Underlying factors (intermediate)** | | | | |
| Water source (n = 19,039) | | | | |
| Improved | 17,769 | 93.8 | 93.4 | 94.2 |
| Not improved | 1,270 | 6.2 | 5.8 | 6.6 |
| Household appliances (n = 19,039) | | | | |
| Owns appliances | 16,796 | 89.9 | 89.5 | 90.4 |
| No appliances | 2,243 | 10.1 | 9,6 | 10.5 |
| Transportation (n = 19039) | | | | |
| Owns transportation | 7,551 | 37.8 | 37.0 | 38.6 |

*(Continued)*

**Table 1.** (Continued)

| Variables | Unweighted absolute frequency | Weighted proportion | 95% confidence interval | |
|---|---|---|---|---|
| | | | LL | UL |
| No transportation | 11,488 | 62.3 | 61.4 | 63.0 |
| Flooring material (n = 18,852) | | | | |
| Improved | 13,738 | 75.3 | 74.7 | 75.9 |
| Not improved | 5,114 | 24.7 | 24.1 | 25.3 |
| Walling material (n = 18,700) | | | | |
| Improved | 18,112 | 97.6 | 97.3 | 97.8 |
| Not improved | 588 | 2.4 | 2.2 | 2.7 |
| Roofing material (n = 18,700) | | | | |
| Improved | 17,868 | 95.9 | 95.6 | 96.2 |
| Not improved | 832 | 4.1 | 3.8 | 4.4 |
| **Proximal factors (immediate)** | | | | |
| Sex | | | | |
| Male | 9,923 | 51.5 | 50.6 | 52.4 |
| Female | 9,268 | 48.5 | 47.6 | 49.4 |
| Child age (months) | | | | |
| 6 a 11 | 2,344 | 12.5 | 11.9 | 13.1 |
| 12 a 23 | 4,063 | 20.7 | 20.0 | 21.4 |
| 24 a 35 | 4,122 | 21.8 | 21.1 | 22.5 |
| 36 a 47 | 4,292 | 22.4 | 21.7 | 23.1 |
| 48 a 59 | 4,370 | 22.7 | 21.9 | 23.4 |
| Birthweight (n = 18640) | | | | |
| Underweight | 1,165 | 6.2 | 5.8 | 6.7 |
| Normal | 15,997 | 85.9 | 85.2 | 86.5 |
| Macrosomic | 1,478 | 7.9 | 7.4 | 8.4 |
| Delivery | | | | |
| Single delivery | 19,010 | 99.0 | 98.8 | 99.1 |
| Multiple birth | 181 | 1.0 | 0.9 | 1.2 |
| Cesarean delivery | | | | |
| No | 12,908 | 64.5 | 63.7 | 65.3 |
| Yes | 6,283 | 35.5 | 34.7 | 36.3 |
| Minimum dietary diversity (n = 12,280) | | | | |
| No | 2,372 | 18.8 | 17.9 | 19.6 |
| Yes | 9,908 | 81.3 | 80.4 | 92.1 |
| Immediate breastfeeding (n = 19,116) | | | | |
| No | 8,996 | 51.1 | 50.2 | 52.0 |
| Yes | 10,120 | 48.9 | 48.0 | 49.8 |

LL = lower limit, UL = upper limit.

Table 2. In certain strata of basic factors, the prevalence of CAS exceeded 10%. Notably, in rural areas, it reached 12.2%; among the poorest, it was 13.4%; among children of mothers with no education and primary education, it registered at 12.6%; for children of mothers with ethnic affiliation, it markedly stood at 17.6%; and among children with low birth weight, the prevalence reached 13.9% (Table 3).

**Table 2. Frequency of concurrence of anemia and stunting in children 6–59 months of age.**

| Outcome | N | Weighted prevalence | 95% confidence interval | |
|---|---|---|---|---|
| | | | LL | UL |
| Anemia | | | | |
| No | 12,230 | 66.0 | 65.2 | 66.8 |
| Yes | 6,961 | 34.0 | 33.1 | 34.8 |
| Stunting | | | | |
| No stunting | 16,706 | 88.4 | 87.9 | 88.9 |
| Moderate stunting | 2,143 | 10.1 | 9.6 | 10.6 |
| Severe stunting | 342 | 1.5 | 1.3 | 1.7 |
| Concurrence of anemia and stunting | | | | |
| No | 17,935 | 93.4 | 94.1 | 94.8 |
| Yes | 1,256 | 5.6 | 5.2 | 5.9 |

N = Unweighted absolute frequency, LL = lower limit, UL = upper limit.

## Associated factors with the concurrence of anemia and stunting

**Bivariate analysis.** Among the distal factors, there was an association of CAS with residence. The maternal employment status was also associated, with a prevalence of 6.1% in households where the mother did not work compared to 5.1% in households where she did work. We found increased prevalence of CAS as the wealth index decreased [13.4% among the poorest and 1.0% among the richest (p < 0.001)]. Women in a relationship had children with higher prevalence of with respect to non-union mothers, there were also differences according to insurance coverage of the mother. A lower educational level of the mother and father were associated with a higher prevalence of CAS. The region was associated with CAS, which was most prevalent in the highlands (9.8%) and jungle (9.1%) of Peru.

All intermediate factors analyzed had a statistical association with CAS; prevalence was higher among households with unimproved materials and lower possessions. Regarding proximal factors, the prevalence of CAS was higher among males; there was also higher prevalence among children aged 12 to 23 months. CAS was associated with the child's birthweight, ranging from 13.9% among those born with underweight and 1.8% in macrosomic children. In those children who did not have minimal dietary diversity there was a higher prevalence of CAS. Children who were immediately breastfed had a higher prevalence of CAS (6.9% versus 4.3%) (Table 3).

**Multivariate analysis.** In model 1 we included those basic factors that were found to be associated at the crude level with CAS. We found that the "poorest" and "poorer" levels increased the odds of CAS compared to the "richest" level. Women with no formal education or with primary level had 2.03 times the odds of children with CAS than children of women with higher education (95%CI: 1.16 to 2.08, p < 0.001), even a dose-response gradient was observed with higher educational level of the mother; a similar association was found with the educational level of the father. Maternal ethnicity increased the odds of a child with CAS by 64% compared to mothers without such ethnicity. Children residing in the highlands and jungle had 2.94 and 2.65 times the odds of CAS compared to those residing in Metropolitan Lima.

In model 2 we included, in addition to the basic factors, the intermediate factors that were significant in the bivariate analysis. Notably, the type of water source and roofing material emerged as associated factors. Children residing in homes with unimproved water sources had 36% higher odds of CAS compared to those with improved water sources (95%CI: 1.10 to 1.68, p = 0.005).

**Table 3. Bivariate analysis to explore factors associated with the concurrence of anemia and stunting in children 6 to 59 months of age.**

| Variables | N | n | Weighted prevalence | Confidence interval 95% | | p-value |
|---|---|---|---|---|---|---|
| | | | | LI | LS | |
| **Basic factors (distal)** | | | | | | |
| Residence | | | | | | |
| Urban | 13,057 | 484 | 3.1 | 2.8 | 3.5 | <0.001 |
| Rural | 6,134 | 772 | 12.2 | 11.3 | 13.2 | |
| Mother currently working | | | | | | |
| No | 8,182 | 579 | 6.1 | 5.6 | 6.8 | 0.007 |
| Yes | 11,009 | 677 | 5.1 | 4.7 | 5.6 | |
| Wealth index | | | | | | |
| Poorest | 5,993 | 844 | 13.4 | 12.5 | 14.4 | <0.001 |
| Poorer | 5,175 | 250 | 4.5 | 3.9 | 5.3 | |
| Medium | 3,778 | 92 | 2.0 | 1.5 | 2.5 | |
| Wealthier | 2,593 | 53 | 1.8 | 1.3 | 2.5 | |
| Wealthiest | 1,652 | 17 | 1.0 | 0.6 | 1.7 | |
| Maternal marital status | | | | | | |
| United | 15,855 | 1072 | 5.8 | 5.4 | 6.2 | 0.004 |
| Not united | 3,336 | 184 | 4.4 | 3.8 | 5.3 | |
| Maternal insurance coverage | | | | | | |
| No | 1,852 | 71 | 3.1 | 2.3 | 4.1 | <0.001 |
| Yes | 17,339 | 1185 | 5.9 | 5.5 | 6.3 | |
| Maternal education | | | | | | |
| No formal education/primary | 3,631 | 485 | 12.6 | 11.4 | 13.9 | <0.001 |
| Secondary | 9,514 | 626 | 5.5 | 5.0 | 6.0 | |
| Higher | 6,046 | 145 | 1.8 | 1.5 | 2.2 | |
| Paternal education (n = 18,304) | | | | | | |
| No formal education/primary | 2,697 | 352 | 12.5 | 11.2 | 14.0 | <0.001 |
| Secondary | 9,702 | 668 | 5.7 | 5.2 | 6.2 | |
| Higher | 5,905 | 174 | 2.4 | 2.0 | 2.8 | |
| Maternal ethnicity | | | | | | |
| No | 17,179 | 876 | 4.6 | 4.2 | 4.9 | <0.001 |
| Yes | 2,012 | 380 | 17.6 | 15.8 | 19.5 | |
| Natural region | | | | | | |
| Metropolitan Lima | 2,310 | 34 | 1.4 | 1.0 | 2.0 | <0.001 |
| Rest of coast | 5,504 | 127 | 2.5 | 2.0 | 3.1 | |
| Highlands | 6,383 | 658 | 9.8 | 9.0 | 10.7 | |
| Jungle | 4,994 | 437 | 9.1 | 8.2 | 10.1 | |
| **Underlying factors (intermediate)** | | | | | | |
| Water source (n = 19,039) | | | | | | |
| Improved | 17,769 | 1,051 | 5.0 | 4.7 | 5.4 | <0.001 |
| Not improved | 1,270 | 197 | 13.8 | 11.9 | 16.0 | |
| Household appliances (n = 19,039) | | | | | | |
| Owns appliances | 16,796 | 934 | 4.7 | 4.4 | 5.1 | <0.001 |
| No appliances | 2,243 | 314 | 13.2 | 11.7 | 14.9 | |
| Transportation (n = 19,039) | | | | | | |
| Owns transportation | 7,551 | 360 | 4.2 | 3.7 | 4.8 | <0.001 |
| No transportation | 11,488 | 888 | 6.4 | 5.9 | 6.9 | |

*(Continued)*

**Table 3.** (*Continued*)

| Variables | N | n | Weighted prevalence | Confidence interval 95% | | p-value |
|---|---|---|---|---|---|---|
| | | | | LI | LS | |
| Flooring material (n = 18,852) | | | | | | |
| Improved | 13,738 | 597 | 3.6 | 3.3 | 4.0 | <0.001 |
| Not improved | 5,114 | 639 | 11.5 | 10.6 | 12.5 | |
| Walling material (n = 18700) | | | | | | |
| Improved | 18,112 | 1139 | 5.4 | 5.0 | 5.7 | <0.001 |
| Not improved | 588 | 89 | 14.0 | 11.1 | 17.6 | |
| Roofing material (n = 18700) | | | | | | |
| Improved | 17,868 | 1118 | 5.4 | 5.0 | 5.7 | <0.001 |
| Not improved | 832 | 110 | 10.7 | 8.7 | 13.1 | |
| **Proximal (immediate) factors** | | | | | | |
| Sex of child | | | | | | |
| Male | 9,923 | 699 | 6.1 | 5.6 | 6.6 | 0.004 |
| Female | 9,268 | 557 | 5.0 | 4.6 | 5.5 | |
| Age of child (months) | | | | | | |
| 6 a 11 | 2,344 | 203 | 7.4 | 6.3 | 8.6 | <0.001 |
| 12 a 23 | 4,063 | 471 | 10.0 | 9.0 | 11.1 | |
| 24 a 35 | 4,122 | 223 | 4.2 | 3.6 | 4.9 | |
| 36 a 47 | 4,292 | 197 | 4.1 | 3.5 | 4.8 | |
| 48 a 59 | 4,370 | 162 | 3.3 | 2.7 | 4.0 | |
| Birthweight (n = 18640) | | | | | | |
| Underweight | 1,165 | 189 | 13.9 | 11.8 | 16.3 | <0.001 |
| Normal | 15,997 | 902 | 4.8 | 4.4 | 5.1 | |
| Macrosomic | 1,478 | 30 | 1.8 | 1.2 | 2.8 | |
| Delivery | | | | | | |
| Single delivery | 19,010 | 1241 | 5.6 | 5.2 | 5.9 | 0.609 |
| Multiple birth | 181 | 15 | 6.5 | 3.6 | 11.3 | |
| Cesarean delivery | | | | | | |
| No | 12,908 | 974 | 6.7 | 6.3 | 7.2 | <0.001 |
| Yes | 6,283 | 282 | 3.5 | 3.0 | 4.0 | |
| Minimum dietary diversity (n = 12,280) | | | | | | |
| No | 2,372 | 266 | 9.5 | 8.3 | 10.9 | <0.001 |
| Yes | 9,908 | 747 | 6.3 | 5.8 | 6.9 | |
| Milk consumption (n = 12,301) | | | | | | |
| No | 4,512 | 473 | 9.3 | 8.4 | 10.3 | <0.001 |
| Yes | 7,789 | 544 | 5.6 | 5.0 | 6.2 | |
| Consumption of tubers and roots (n = 12,294) | | | | | | |
| No | 3,460 | 286 | 7.1 | 6.2 | 8.1 | 0.610 |
| Yes | 8,834 | 729 | 6.8 | 6.3 | 7.4 | |
| Egg consumption (n = 12,290) | | | | | | |
| No | 5,634 | 526 | 8.0 | 7.2 | 8.8 | <0.001 |
| Yes | 6,656 | 488 | 6.1 | 5.5 | 6.7 | |
| Meat consumption (n = 12,291) | | | | | | |
| No | 2,589 | 307 | 10.9 | 9.6 | 12.4 | <0.001 |
| Yes | 9,702 | 709 | 5.9 | 5.4 | 6.4 | |
| Consumption of fruits and vegetables rich in vitamin A (n = 12,298) | | | | | | |

(*Continued*)

**Table 3.** (Continued)

| Variables | N | n | Weighted prevalence | Confidence interval 95% | | p-value |
|---|---|---|---|---|---|---|
| | | | | LI | LS | |
| No | 3,163 | 328 | 8.9 | 7.9 | 10.1 | <0.001 |
| Yes | 9,135 | 688 | 6.3 | 5.7 | 6.8 | |
| Consumption of other fruits and vegetables (n = 12,297) | | | | | | |
| No | 2,198 | 203 | 8.5 | 7.3 | 9.9 | 0.005 |
| Yes | 10,099 | 813 | 6.6 | 6.1 | 7.2 | |
| Consumption of legumes and dried fruits and nuts (n = 12,297) | | | | | | |
| No | 6,441 | 575 | 7.3 | 6.6 | 8.0 | 0.156 |
| Yes | 5,856 | 441 | 6.5 | 5.9 | 7.3 | |
| Immediate breastfeeding (n = 19,116) | | | | | | |
| No | 8,996 | 461 | 4.3 | 3.9 | 4.8 | <0.001 |
| Yes | 10,120 | 791 | 6.9 | 6.3 | 7.5 | |

N = Unweighted absolute frequency, n = unweighted count of concurrence of anemia and stunting, LL = lower limit, UL = upper limit.

In model 3, we extended our analysis to include proximal factors in addition to the previously considered factors. Interestingly, all the basic factors retained their association with CAS, whereas none of the intermediate factors showed significant associations. Male sex, age groups 6 to 11 months and 12 to 23 months, and low birth weight independently increased the odds of CAS. Boys had 27% higher odds compared to girls (95%CI: 1.05 to 1.52, p = 0.012), children aged 12 to 23 months had 2.89 times the odds of CAS compared to those aged 48 to 59 months. The highest risk was among those with low birth weight (aOR: 7.31, 95%CI: 4.26 to 12.54) compared to macrosomic (aOR: 2.07, 95%CI: 1.28 to 3.37) (Table 4).

## Discussion

In this study, the prevalence of CAS in children aged 6–59 months was 5.6%, a similar result to other lower-middle-income countries such as Armenia (4.37%), Egypt (6.44%) [26] and Ghana (12%) [8]. In the systematic review by Tran. T et al. higher prevalences were identified in low-income countries such as Nigeria (36.71%), India (38.38%) and Yemen (43.34%). On the other hand, other lower-middle-income countries had lower prevalences, such as Moldova (3.44%), Jordan (3.29%) and Albania (2.57%) [24].

CAS is a complex phenomenon in which its component disorders interact negatively in the body. Stunting affects the production of red blood cells and hemoglobin through mechanisms such as decreased erythropoietin production due to the action of proinflammatory cytokines, and deficiency in the intake of essential nutrients such as iron, vitamins and minerals, exacerbating the onset of anemia [27]. Also, anemia reduces oxygen-carrying capacity, aggravating underlying stunting or giving rise to it, and compromising physical and cognitive development [28].

CAS in children is associated with an increased risk of adverse outcomes related to school performance, future work capacity, as well as greater costs and economic losses to the family and society compared to those with only one of these conditions [28]. This synergy is detrimental and although some studies mention that the repercussions generated by each entity are independent, it is plausible that the coexistence represents a significant threat to the health and integral development of children [29].

**Table 4. Hierarchical binary logistic regression of factors associated with the concurrence of anemia and stunting in children 6 to 59 months of age.**

| Variables | Model 1 [a] | | | | Model 2 [b] | | | | Model 3 [c] | | | |
|---|---|---|---|---|---|---|---|---|---|---|---|---|
| | aOR | LL | UL | p-value | aOR | LL | UL | p-value | aOR | LL | UL | p-value |
| **Basic factors (distal)** | | | | | | | | | | | | |
| Residence | | | | | | | | | | | | |
| Urban | 1.03 | 0.85 | 1.26 | 0.735 | 1.08 | 0.88 | 1.32 | 0.471 | 1.12 | 0.88 | 1.43 | 0.369 |
| Rural | 1 | | | | 1 | | | | 1 | | | |
| Mother currently working | | | | | | | | | | | | |
| No | 1.11 | 0.95 | 1.29 | 0.186 | 1.09 | 0.93 | 1.27 | 0.269 | 0.88 | 0.73 | 1.07 | 0.213 |
| Yes | 1 | | | | 1 | | | | 1 | | | |
| Wealth index | | | | | | | | | | | | |
| Poorest | 3.87 | 1.99 | 7.50 | <0.001 | 3.39 | 1.65 | 6.96 | 0.001 | 3.72 | 1.66 | 8.31 | 0.001 |
| Poorer | 2.07 | 1.08 | 3.98 | 0.028 | 2.09 | 1.04 | 4.20 | 0.039 | 2.17 | 1.00 | 4.72 | 0.050 |
| Medium | 1.17 | 0.61 | 2.25 | 0.642 | 1.23 | 0.61 | 2.47 | 0.565 | 1.30 | 0.60 | 2.82 | 0.512 |
| Wealthier | 1.49 | 0.76 | 2.93 | 0.247 | 1.51 | 0.74 | 3.10 | 0.260 | 1.22 | 0.54 | 2.75 | 0.638 |
| Wealthiest | 1 | | | | 1 | | | | 1 | | | |
| Mother's marital status | | | | | | | | | | | | |
| United | 1.14 | 0.90 | 1.44 | 0.274 | 1.16 | 0.91 | 1.47 | 0.239 | 0.96 | 0.71 | 1.28 | 0.775 |
| Not united | 1 | | | | 1 | | | | 1 | | | |
| Maternal insurance coverage | | | | | | | | | | | | |
| No | 1 | | | | 1 | | | | 1 | | | |
| Yes | 1.15 | 0.85 | 1.57 | 0.361 | 1.15 | 0.84 | 1.57 | 0.378 | 1.16 | 0.78 | 1.73 | 0.462 |
| Maternal education | | | | | | | | | | | | |
| No formal education/ primary | 2.03 | 1.46 | 2.81 | <0.001 | 2.01 | 1.44 | 2.81 | <0.001 | 2.10 | 1.43 | 3.07 | <0.001 |
| Secondary | 1.55 | 1.16 | 2.08 | 0.003 | 1.56 | 1.16 | 2.11 | 0.004 | 1.54 | 1.10 | 2.16 | 0.013 |
| Higher | 1 | | | | 1 | | | | 1 | | | |
| Paternal education (n = 18304) | | | | | | | | | | | | |
| No formal education/primary | 1.55 | 1.16 | 2.07 | 0.003 | 1.49 | 1.11 | 2.00 | 0.007 | 1.57 | 1.13 | 2.19 | 0.008 |
| Secondary | 1.22 | 0.95 | 1.57 | 0.112 | 1.19 | 0.93 | 1.53 | 0.162 | 1.16 | 0.88 | 1.53 | 0.298 |
| Higher | 1 | | | | 1 | | | | 1 | | | |
| Mother ethnicity | | | | | | | | | | | | |
| No | 1 | | | | 1 | | | | 1 | | | |
| Yes | 1.64 | 1.38 | 1.96 | <0.001 | 1.53 | 1.27 | 1.84 | <0.001 | 1.30 | 1.02 | 1.66 | 0.033 |
| Natural region | | | | | | | | | | | | |
| Metropolitan Lima | 1 | | | | 1 | | | | 1 | | | |
| Rest of the coast | 1.29 | 0.82 | 2.03 | 0.264 | 1.19 | 0.74 | 1.91 | 0.479 | 1.45 | 0.82 | 2.57 | 0.200 |
| Highlands | 2.94 | 1.90 | 4.53 | <0.001 | 3.28 | 2.08 | 5.17 | <0.001 | 3.91 | 2.23 | 6.83 | <0.001 |
| Jungle | 2.65 | 1.71 | 4.09 | <0.001 | 2.68 | 1.70 | 4.25 | <0.001 | 2.87 | 1.63 | 5.05 | <0.001 |
| **Underlying factors (intermediate)** | | | | | | | | | | | | |
| Water source | | | | | | | | | | | | |
| Improved | | | | | 1 | | | | 1 | | | |
| Not improved | | | | | 1.36 | 1.10 | 1.68 | 0.005 | 1.28 | 0.96 | 1.71 | 0.091 |
| Household appliances | | | | | | | | | | | | |
| Owns appliances | | | | | 1 | | | | 1 | | | |
| No appliances | | | | | 1.10 | 0.92 | 1.33 | 0.297 | 0.99 | 0.78 | 1.26 | 0.94 |
| Transportation | | | | | | | | | | | | |
| Owns transportation | | | | | 1 | | | | 1 | | | |
| No transportation | | | | | 1.10 | 0.93 | 1.30 | 0.271 | 1.04 | 0.85 | 1.27 | 0.71 |
| Flooring material | | | | | | | | | | | | |

*(Continued)*

**Table 4.** (Continued)

| Variables | Model 1 [a] | | | | Model 2 [b] | | | | Model 3 [c] | | | |
|---|---|---|---|---|---|---|---|---|---|---|---|---|
| | aOR | LL | UL | p-value | aOR | LL | UL | p-value | aOR | LL | UL | p-value |
| Improved | | | | | 1 | | | | 1 | | | |
| Not improved | | | | | 1.10 | 0.92 | 1.31 | 0.299 | 0.98 | 0.79 | 1.23 | 0.877 |
| Walling material | | | | | | | | | | | | |
| Improved | | | | | 1 | | | | 1 | | | |
| Not improved | | | | | 1.34 | 0.97 | 1.86 | 0.076 | 1.36 | 0.87 | 2.13 | 0.184 |
| Roofing material | | | | | | | | | | | | |
| Improved | | | | | 1 | | | | 1 | | | |
| Not improved | | | | | 1.49 | 1.12 | 1.98 | 0.007 | 1.27 | 0.86 | 1.87 | 0.23 |
| **Proximal factors (immediate)** | | | | | | | | | | | | |
| Sex | | | | | | | | | | | | |
| Male | | | | | | | | | 1.27 | 1.05 | 1.52 | 0.012 |
| Female | | | | | | | | | 1 | | | |
| Age of child (months) | | | | | | | | | | | | |
| 6 a 11 | | | | | | | | | 1.89 | 1.23 | 2.91 | 0.004 |
| 12 a 23 | | | | | | | | | 2,89 | 1.93 | 4.35 | <0.001 |
| 24 a 35 | | | | | | | | | 1.16 | 0.75 | 1.77 | 0.509 |
| 36 a 47 | | | | | | | | | 1.14 | 0.68 | 1.93 | 0.612 |
| 48 a 59 | | | | | | | | | 1 | | | |
| Birthweight | | | | | | | | | | | | |
| Underweight | | | | | | | | | 7.31 | 4.26 | 12.54 | <0.001 |
| Normal | | | | | | | | | 2.07 | 1.28 | 3.37 | 0.003 |
| Macrosomic | | | | | | | | | 1 | | | |
| Delivery | | | | | | | | | | | | |
| Single delivery | | | | | | | | | 1 | | | |
| Multiple birth | | | | | | | | | 1.11 | 0.86 | 1.42 | 0.414 |
| Cesarean delivery | | | | | | | | | | | | |
| No | | | | | | | | | 1.03 | 0.82 | 1.30 | 0.801 |
| Yes | | | | | | | | | | | | |
| Immediate breastfeeding | | | | | | | | | | | | |
| No | | | | | | | | | 1.11 | 0.89 | 1.37 | 0.362 |
| Yes | | | | | | | | | 1 | | | |

aOR = adjusted odds ratio, LL = lower limit, UL = upper limit

Model 1: adjusted for the variables residence, mother currently working, wealth index, maternal marital status, maternal insurance coverage, maternal education, paternal education, maternal ethnicity, natural region.

Model 2: adjusted for all variables in model 1 and the variables water source, possession of household appliances, transportation, flooring material, walling material and roofing material.

Model 3: adjusted for all variables in model 1 and 2, and the variables sex, age of child, birthweight, cesarean delivery, minimum dietary diversity, and immediate breastfeeding.

[a] Pseudo R2 Nagelkerke R2 = 0.155, McFadden's R2 = 0.129. Standard errors of the coefficients were between 0.079 and 0.377.

[b] Pseudo R2 Nagelkerke = 0.163, McFadden's R2 = 0.136. The standard errors of the coefficients were between 0.080 and 0.410.

[c] Pseudo R2 Nagelkerke = 0.204, R2 McFadden's = 0.169. The standard errors of the coefficients were between 0.094 and 0.406.

Among the associated factors, we observed a dose-response relationship with the wealth index. Children from the poorest and poorer quintiles exhibited higher odds of CAS compared to those with a wealthier index. This observed trend persisted consistently across all three models, implying that the wealth index influences the incidence of CAS through mechanisms

beyond the intermediate and proximal factors considered in this study. The wealth index of a household is known to affect the ability of its members to access health care services; the quality of their nutrition; as well as hygiene and sanitation conditions, which may reduce the risk of water- and vector-borne diseases [30, 31]. Previous studies are consistent in reporting that the odds of CAS were higher in children located in the lower wealth quintiles [7, 8, 22].

Children with mothers who had no formal education or only had primary education had twice the odds of suffering from CAS compared to children with mothers who had higher education. Furthermore, we found a dose-response relationship that strengthens the causal association between the mother's educational level and the development of CAS. This finding is consistent with other studies that demonstrate that in lower education categories, the risk of CAS is increased [7, 8, 31, 32].

The same trend, although with a lower strength of association, is observed with respect to paternal education. This factor has been previously noted as associated in another study [32]. While maternal education appears to exert a stronger influence on CAS due to its central role in childcare and feeding, paternal educational level nonetheless emerges as a risk factor for this condition. This finding remained consistent across all three models, underscoring the adverse impact of fathers having no formal education or only completing primary school.

Maternal ethnicity, constructed based on ethnolinguistic family grouping, is associated with a higher risk of CAS compared to those whose mothers speak Spanish, English or Portuguese. Even in models that include intermediate and basic factors, we observed that the strength of association is still present and significant. A study in a sample of 688 Peruvian children aged 6 to 36 months found that the CAS are associated with having a Quechua-speaking mother [33]. This association is rooted in the fact that in Peru, the Quechua-speaking population exhibits poorer socioeconomic indicators, which impact their ability to access the health system and compromise their food security.

We found that the prevalence of CAS was significantly higher among children in the highlands and jungle of Peru, compared to those residing in the capital of the country, mainly in the urban area. The increased prevalence of CAS in these regions and the magnitudes of the estimated odds ratio, even in models that include intermediate and proximal factors, would reflect a direct effect not mediated by characteristics more proximal to the child. The geographic and environmental conditions in these regions play a pivotal role in shaping the availability of and access to nutritious food, water quality, and medical care, consequently elevating the risk to children's health. Moreover, variations in diets and eating patterns may contribute to a higher prevalence of diets deficient in essential nutrients, thereby increasing the risk of CAS [34]. Simultaneously, socioeconomic inequalities between these regions also impact quality of life, access to health services, and the ability of families to provide adequate nutrition for their children [23].

Children living in households with unimproved water sources had 36% higher odds of CAS compared to those with access to improved sources. This variable acts independently of the confounding effect generated by variables such as wealth index, parental education level, ethnicity, and natural region. This finding aligns with prior studies demonstrating that enhanced access to safe and potable water at home reduces the incidence of diseases that can lead to anemia, stunting, or both concurrently [33, 35].

We can see a similar relationship with the roofing material of the dwelling. Children residing in homes with thatched roofs have been shown to be twice as likely to experience episodes of diarrhea compared to those living in homes covered with corrugated iron roofs [36]. In turn, children living in homes with rudimentary roofing materials are 1.11 times more likely to suffer from diarrheal disease than those children in homes with concrete roofing materials [37], and those living in thatched roof homes are 8% more likely to experience diarrheal

disease than those children in homes with metal or concrete roofs [38]. These studies identified that the type of housing roofing material is related to an increased risk of infections, and that these poor housing conditions may contribute to and explain an increased risk of CAS in resident children.

The results revealed that boys had 27% higher odds of CAS compared to girls. These data are consistent with a previous study, where 25% higher odds of CAS were observed among boys [21]. From a biological perspective, boys are often more susceptible to stunting than girls due to differences in body composition, having higher fat-free mass and lower fat mass compared to girls, resulting in higher energy demands to maintain adequate growth and development [39]. Also, hypothalamic-pituitary-gonadal axis hormones (testosterone, early disappearance of luteinizing hormone and follicle-stimulating hormone) increase the susceptibility of boys for stunting compared to girls [39, 40].

In terms of age, children aged 12 to 23 months had 2.89 times the odds of CAS compared to children aged 48 to 59 months. A dose-response relationship was evident, where at younger ages, the risk of CAS was higher. These results agree with previous research that has reported a higher prevalence of CAS in children younger than 2 years, especially in developing countries [21, 22, 41]. In this study, the highest risk was observed in the 12- to 23-month-old group, and as children grew older, the risk decreased compared with children aged 48 to 59 months.

This observation could be explained by the transition process of children's diets from a predominantly liquid diet (such as exclusive breastfeeding) to a solid diet. During this period, nutritional requirements undergo significant changes, and dietary imbalances or late introduction of complementary foods may affect the child's health. These findings are consistent with research that has pointed out that the progressive decrease in iron stores in the body, especially between 4 and 6 months, together with inadequate diets and susceptibility to infections, contribute to the increased risk of CAS at this stage of infancy [32, 42, 43].

We found that children with low birth weight had 7.31 times the odds of CAS compared to macrosomic children. Low birthweight is known to be associated with deficiency of nutritional factors, such as health care [44, 45]. Low birthweight infants have often experienced poor fetal nutrition, which can result in poor organ and tissue development, leading to an increased predisposition to anemia and an inability to effectively absorb and utilize nutrients, contributing to stunting. In addition, these children may have less developed immune systems, making them more susceptible to infections that can lead to anemia and worsen stunting [42].

The lack of association between dietary diversity and CAS in our study reflects the complexity of this condition and suggests that several factors may be influencing its development. Dietary diversity is only one of many components related to children's nutrition and health. Other factors, such as the availability of nutritious foods, access to health care services, as well as the incidence of parasitic infections or chronic inflammatory states in children, could play a more relevant role in CAS. In addition, geographic and cultural differences may also influence how dietary diversity is assessed. The lack of association does not rule out the importance of a balanced diet, but highlights that CAS is a multifactorial condition and that a comprehensive approach addressing various determinants is essential to prevent and control this problem [46].

One of the strengths of this study is that it was based on the DHS 2022, which due to its complex sample design is nationally representative. In addition, it was conducted under a standardized methodology of demographic and health surveys, which allows for some comparisons. Another strength is that our study is the first done specifically to assess the prevalence of CAS in Peru, as well as to identify relevant factors within the UNICEF conceptual framework on the determinants of maternal and child nutrition.

The study is not exempt from limitations, and these should be considered when interpreting the results. Firstly, the cross-sectional design prevents the establishment of a temporal sequence in the identified associations. The second limitation pertains to the measurement of variables; certain data on the children were reliant on the mother's report, and the extended recall period may introduce social desirability and recall bias, respectively. Lastly, the study did not encompass all potential covariates that could have a relevant association with our variable of interest.

In conclusion, in Peru we found that the concurrence of anemia and stunting affected 5.6% of children aged 6 to 59 months. The prevalence of this condition was unevenly distributed, which allowed us to identify a series of factors that increased the risk of CAS. Among the distal factors were lower wealth index quintiles, lower educational level of the mother and father, belonging to an ethnic group, and residing in the jungle and highlands. Intermediate factors included having an unimproved water source and unimproved roofing material; and proximal factors included being a male child, being 12 to 23 months old, and being born with low birth weight. These findings allow us to identify determinants, including several modifiable factors, that could be addressed with specific interventions and in different scenarios, from prenatal control to improving housing conditions.

## Supporting information

**S1 Table. Variable definitions.**
(DOCX)

**S1 Data. Raw data.**
(DTA)

## Author Contributions

**Conceptualization:** Alessandra Rivera, Víctor Marín.

**Data curation:** Alessandra Rivera, Víctor Marín, Franco Romaní.

**Formal analysis:** Alessandra Rivera, Víctor Marín, Franco Romaní.

**Funding acquisition:** Alessandra Rivera, Víctor Marín.

**Investigation:** Alessandra Rivera, Víctor Marín, Franco Romaní.

**Methodology:** Alessandra Rivera, Víctor Marín, Franco Romaní.

**Project administration:** Alessandra Rivera, Víctor Marín.

**Resources:** Alessandra Rivera, Víctor Marín.

**Software:** Alessandra Rivera, Víctor Marín, Franco Romaní.

**Supervision:** Alessandra Rivera, Víctor Marín, Franco Romaní.

**Validation:** Alessandra Rivera, Víctor Marín, Franco Romaní.

**Visualization:** Alessandra Rivera, Víctor Marín, Franco Romaní.

**Writing – original draft:** Alessandra Rivera, Víctor Marín, Franco Romaní.

**Writing – review & editing:** Alessandra Rivera, Víctor Marín, Franco Romaní.

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
