## [Decision Letter · Decision Letter 0]

28 Dec 2023

PGPH-D-23-02302

Concurrence of anemia and stunting in children aged 6 to 59 months and its associated factors in Peru

Dear Dr. Marin,

Thank you for submitting your manuscript to PLOS Global Public Health. After careful consideration, we feel that it has merit but does not fully meet PLOS Global Public Health’s publication criteria as it currently stands. Therefore, we invite you to submit a revised version of the manuscript that addresses the points raised during the review process.

We look forward to receiving your revised manuscript.

Kind regards,

Dessalegn Tamiru

Academic Editor

Journal Requirements:

Additional Editor Comments (if provided):

(Reviewer 1)

Reviewer Recommendation Term: Minor Revision

Rate Review: 0

Custom Review Question(s): Response

Comments to the Author

1. Does this manuscript meet PLOS Global Public Health’s publication criteria? Is the manuscript technically sound, and do the data support the conclusions? The manuscript must describe methodologically and ethically rigorous research with conclusions that are appropriately drawn based on the data presented. Yes

2. Has the statistical analysis been performed appropriately and rigorously? Yes

3. Have the authors made all data underlying the findings in their manuscript fully available (please refer to the Data Availability Statement at the start of the manuscript PDF file)?

The PLOS Data policy requires authors to make all data underlying the findings described in their manuscript fully available without restriction, with rare exception. The data should be provided as part of the manuscript or its supporting information, or deposited to a public repository. For example, in addition to summary statistics, the data points behind means, medians and variance measures should be available. If there are restrictions on publicly sharing data—e.g. participant privacy or use of data from a third party—those must be specified. Yes

4. Is the manuscript presented in an intelligible fashion and written in standard English?

PLOS Global Public Health does not copyedit accepted manuscripts, so the language in submitted articles must be clear, correct, and unambiguous. Any typographical or grammatical errors should be corrected at revision, so please note any specific errors here. Yes

5. Review Comments to the Author

Please use the space provided to explain your answers to the questions above. You may also include additional comments for the author, including concerns about dual publication, research ethics, or publication ethics. (Please upload your review as an attachment if it exceeds 20,000 characters) Review report

Title: Concurrence of anemia and stunting in children aged 6 to 59 months and its associated factors in Peru

Dear Editor:

Thank you very much for inviting me to review an article entitled “Concurrence of anemia and stunting in children aged 6 to 59 months and its associated factors in Peru”.

The study is interesting in its field considering the magnitude of the problem globally, and the study fills the knowledge gap on concurrent anemia and stunting in the country by using a nationally representative data.

The paper is well written. However, the authors need to consider the following major and minor concern substantially.

Abstract section

1. Line 18 -19 – the authors stated “The analysis design was cross-sectional….... better to say study design….

2. Procedures for anthropometry and hemoglobin measurement were not described in the abstract section. Pleas describe briefly.

Background

3. The paper provides very limited background information particularly with regards to the efforts made by concerned body to address the issue and the research gap they have intended to fill.

Methods

4. It would be interesting to have some details of the respondents: who were selected to be the responsible for the answers; were the respondents only mothers or any person present at the household could answer the questionnaire? What specific criteria were used to select the respondents?

5. The authors stated that "Variables with P<0.10 value in the bivariate analyses were included in the multivariate analyses. Thus, model-1 included the distal factors which demonstrated P<0.10". Did you consider the theoretical model in the inclusion of variables? It is quite important to understand that the theoretical model should be above statistical significance.

6. The authors built single construct of Concurrent Anemia and Stunting. Have you thought of considering testing a construct of determination of the outcome divided in four categories: only stunting, only anemia, CAS versus none? It would have been interesting to analyze and compare the single odds ratio of each outcome and both outcome to verify if the odds are higher when they are combined then only anemia or stunting.

6. PLOS authors have the option to publish the peer review history of their article (what does this mean?). If published, this will include your full peer review and any attached files.

Do you want your identity to be public for this peer review? For information about this choice, including consent withdrawal, please see our Privacy Policy. Yes:

Confidential to Editor

1. Do you have any potential or perceived competing interests that may influence your review? Please review our Competing Interests policy and declare any potential interests. If you have no competing interests, please write "I have no competing interests." I have no competing interests

2. Did you receive any assistance in preparing this review (e.g. from a post-doc or graduate student)? If yes, please include their name below. (optional)

3. If accepted, do you think this submission should be highlighted on the journal website and/or to the media? (optional)

Do you want to get recognition for this review on a Web of Science researcher profile?

If you opt in, your Web of Science profile will automatically be updated to show a verified record of this review in full compliance with the journal’s review policy. If you don’t have a Web of Science profile, you will be prompted to create a free account.

Yes

(Reviewer 2)

Reviewer Recommendation Term: Minor Revision

Rate Review: 0

Custom Review Question(s): Response

Comments to the Author

1. Does this manuscript meet PLOS Global Public Health’s publication criteria? Is the manuscript technically sound, and do the data support the conclusions? The manuscript must describe methodologically and ethically rigorous research with conclusions that are appropriately drawn based on the data presented. Partly

2. Has the statistical analysis been performed appropriately and rigorously? Yes

3. Have the authors made all data underlying the findings in their manuscript fully available (please refer to the Data Availability Statement at the start of the manuscript PDF file)?

The PLOS Data policy requires authors to make all data underlying the findings described in their manuscript fully available without restriction, with rare exception. The data should be provided as part of the manuscript or its supporting information, or deposited to a public repository. For example, in addition to summary statistics, the data points behind means, medians and variance measures should be available. If there are restrictions on publicly sharing data—e.g. participant privacy or use of data from a third party—those must be specified. Yes

4. Is the manuscript presented in an intelligible fashion and written in standard English?

PLOS Global Public Health does not copyedit accepted manuscripts, so the language in submitted articles must be clear, correct, and unambiguous. Any typographical or grammatical errors should be corrected at revision, so please note any specific errors here. Yes

5. Review Comments to the Author

Please use the space provided to explain your answers to the questions above. You may also include additional comments for the author, including concerns about dual publication, research ethics, or publication ethics. (Please upload your review as an attachment if it exceeds 20,000 characters) Concurrence of anemia and stunting in children aged 6 to 59 months and its associated factors in Peru

Thank you for your contribution. This is a really interesting study that collects information that can be helpful in creating regional and overall plans to implement EBP in healthcare. I'd just like to draw attention to a few problems with certain kinds in the text:

1. Title: Modify as: Concurrence of anemia and stunting and associated factors among children aged 6 to 59 months in Peru

2. Lines # 23 and 24: bivariate analysis and multivariate analysis make it bivariable analysis and multivariable analysis. And also in entire your document.

3. Line #24, you said, “We applied a statistical criterion (p < 0.10 in the crude analysis)." Why did you use p<0.1? Why don't you use 0.2, 0.25, or 0.3?

4. To determine the height-for-age Z-score of the under-5 children, which software was used?

5. Lines 160 and 161 were categorized as not stunting (HW70 ≥ -2 SD), moderate stunting (-3 SD ≥ HW70 < -2 SD), and severe stunting (HW70 < -3 SD). Since you used binary logistic regression, it should be recategorized as a dichotomous variable.

6. Explain how Wealth index was analyzed

7. What is minimum dietary diversity according to this study?

8. Line # 185: Table 1: Variable Definitions What is the importance of this table? If needed, you can submit it as supporting material alone.

9. The technique or method used to determine the concurrent of anemia and stunting is not clearly stated. Please state it clearly under data measurement or data analysis.

10. Line #277-297: It is about bivariable analysis and table 4. What is the importance of describing bivariables in detail? It is better if you mention only the potential candidate of the variable that was selected for multivariable logistic regression.

11. Line # 302-310: Do you think that cured-level association is reported as a statistically associated variable?

6. PLOS authors have the option to publish the peer review history of their article (what does this mean?). If published, this will include your full peer review and any attached files.

Do you want your identity to be public for this peer review? For information about this choice, including consent withdrawal, please see our Privacy Policy. No

Confidential to Editor

1. Do you have any potential or perceived competing interests that may influence your review? Please review our Competing Interests policy and declare any potential interests. If you have no competing interests, please write "I have no competing interests." The manuscript is too large—about 41 pages. It is better if it is reduced.

2. Did you receive any assistance in preparing this review (e.g. from a post-doc or graduate student)? If yes, please include their name below. (optional) No

3. If accepted, do you think this submission should be highlighted on the journal website and/or to the media? (optional) Yes, on the journal website only

Do you want to get recognition for this review on a Web of Science researcher profile?

If you opt in, your Web of Science profile will automatically be updated to show a verified record of this review in full compliance with the journal’s review policy. If you don’t have a Web of Science profile, you will be prompted to create a free account.

Yes

Reviewers' comments:

Reviewer's Responses to Questions

**Comments to the Author**

1. Does this manuscript meet PLOS Global Public Health’s publication criteria? Is the manuscript technically sound, and do the data support the conclusions? The manuscript must describe methodologically and ethically rigorous research with conclusions that are appropriately drawn based on the data presented.

Reviewer #1: Yes

Reviewer #2: Partly

2. Has the statistical analysis been performed appropriately and rigorously?

Reviewer #1: Yes

Reviewer #2: Yes

3. Have the authors made all data underlying the findings in their manuscript fully available (please refer to the Data Availability Statement at the start of the manuscript PDF file)?

Reviewer #1: Yes

Reviewer #2: Yes

4. Is the manuscript presented in an intelligible fashion and written in standard English?

Reviewer #1: Yes

Reviewer #2: Yes

5. Review Comments to the Author

Reviewer #1: Review report

Title: Concurrence of anemia and stunting in children aged 6 to 59 months and its associated factors in Peru

Dear Editor:

Thank you very much for inviting me to review an article entitled “Concurrence of anemia and stunting in children aged 6 to 59 months and its associated factors in Peru”.

The study is interesting in its field considering the magnitude of the problem globally, and the study fills the knowledge gap on concurrent anemia and stunting in the country by using a nationally representative data.

The paper is well written. However, the authors need to consider the following major and minor concern substantially.

Abstract section

1. Line 18 -19 – the authors stated “The analysis design was cross-sectional….... better to say study design….

2. Procedures for anthropometry and hemoglobin measurement were not described in the abstract section. Pleas describe briefly.

Background

3. The paper provides very limited background information particularly with regards to the efforts made by concerned body to address the issue and the research gap they have intended to fill.

Methods

4. It would be interesting to have some details of the respondents: who were selected to be the responsible for the answers; were the respondents only mothers or any person present at the household could answer the questionnaire? What specific criteria were used to select the respondents?

5. The authors stated that "Variables with P<0.10 value in the bivariate analyses were included in the multivariate analyses. Thus, model-1 included the distal factors which demonstrated P<0.10". Did you consider the theoretical model in the inclusion of variables? It is quite important to understand that the theoretical model should be above statistical significance.

6. The authors built single construct of Concurrent Anemia and Stunting. Have you thought of considering testing a construct of determination of the outcome divided in four categories: only stunting, only anemia, CAS versus none? It would have been interesting to analyze and compare the single odds ratio of each outcome and both outcome to verify if the odds are higher when they are combined then only anemia or stunting.

Reviewer #2: Concurrence of anemia and stunting in children aged 6 to 59 months and its associated factors in Peru

Thank you for your contribution. This is a really interesting study that collects information that can be helpful in creating regional and overall plans to implement EBP in healthcare. I'd just like to draw attention to a few problems with certain kinds in the text:

1. Title: Modify as: Concurrence of anemia and stunting and associated factors among children aged 6 to 59 months in Peru

2. Lines # 23 and 24: bivariate analysis and multivariate analysis make it bivariable analysis and multivariable analysis. And also in entire your document.

3. Line #24, you said, “We applied a statistical criterion (p < 0.10 in the crude analysis)." Why did you use p<0.1? Why don't you use 0.2, 0.25, or 0.3?

4. To determine the height-for-age Z-score of the under-5 children, which software was used?

5. Lines 160 and 161 were categorized as not stunting (HW70 ≥ -2 SD), moderate stunting (-3 SD ≥ HW70 < -2 SD), and severe stunting (HW70 < -3 SD). Since you used binary logistic regression, it should be recategorized as a dichotomous variable.

6. Explain how Wealth index was analyzed

7. What is minimum dietary diversity according to this study?

8. Line # 185: Table 1: Variable Definitions What is the importance of this table? If needed, you can submit it as supporting material alone.

9. The technique or method used to determine the concurrent of anemia and stunting is not clearly stated. Please state it clearly under data measurement or data analysis.

10. Line #277-297: It is about bivariable analysis and table 4. What is the importance of describing bivariables in detail? It is better if you mention only the potential candidate of the variable that was selected for multivariable logistic regression.

11. Line # 302-310: Do you think that cured-level association is reported as a statistically associated variable?

6. PLOS authors have the option to publish the peer review history of their article (what does this mean?). If published, this will include your full peer review and any attached files.

**Do you want your identity to be public for this peer review?** **********

---

## [Decision Letter · Decision Letter 1]

12 Mar 2024

Concurrence of anemia and stunting and associated factors among children aged 6 to 59 months in Peru

PGPH-D-23-02302R1

Dear Sr. Marin,

We are pleased to inform you that your manuscript 'Concurrence of anemia and stunting and associated factors among children aged 6 to 59 months in Peru' has been provisionally accepted for publication in PLOS Global Public Health.

Best regards,

Dessalegn Tamiru

Academic Editor

Reviewer Comments (if any, and for reference):

Reviewer's Responses to Questions

**Comments to the Author**

1. If the authors have adequately addressed your comments raised in a previous round of review and you feel that this manuscript is now acceptable for publication, you may indicate that here to bypass the “Comments to the Author” section, enter your conflict of interest statement in the “Confidential to Editor” section, and submit your "Accept" recommendation.

Reviewer #1: All comments have been addressed

Reviewer #2: All comments have been addressed

2. Does this manuscript meet PLOS Global Public Health’s publication criteria? Is the manuscript technically sound, and do the data support the conclusions? The manuscript must describe methodologically and ethically rigorous research with conclusions that are appropriately drawn based on the data presented.

Reviewer #1: Yes

Reviewer #2: Yes

3. Has the statistical analysis been performed appropriately and rigorously?

Reviewer #1: Yes

Reviewer #2: Yes

4. Have the authors made all data underlying the findings in their manuscript fully available (please refer to the Data Availability Statement at the start of the manuscript PDF file)?

Reviewer #1: Yes

Reviewer #2: Yes

5. Is the manuscript presented in an intelligible fashion and written in standard English?

Reviewer #1: Yes

Reviewer #2: Yes

6. Review Comments to the Author

Reviewer #1: (No Response)

Reviewer #2: All the given comments have been addressed

7. PLOS authors have the option to publish the peer review history of their article (what does this mean?). If published, this will include your full peer review and any attached files.

**Do you want your identity to be public for this peer review?** For information about this choice, including consent withdrawal, please see our Privacy Policy.

Reviewer #1: **Yes: **Dr. Dereje Tsegaye Hawetu

Reviewer #2: No
